# Bad Neighborhood: Fibrotic Stroma as a New Player in Melanoma Resistance to Targeted Therapies

**DOI:** 10.3390/cancers12061364

**Published:** 2020-05-26

**Authors:** Serena Diazzi, Sophie Tartare-Deckert, Marcel Deckert

**Affiliations:** 1C3M, Université Côte d’Azur, INSERM, 06204 Nice, France; Serena.DIAZZI@unice.fr; 2Equipe labellisée Ligue Contre le Cancer 2016, 06204 Nice, France

**Keywords:** melanoma, fibrosis, targeted therapies, resistance

## Abstract

Current treatments for metastatic cutaneous melanoma include immunotherapies and drugs targeting key molecules of the mitogen-activated protein kinase (MAPK) pathway, which is often activated by *BRAF* driver mutations. Overall responses from patients with metastatic *BRAF* mutant melanoma are better with therapies combining BRAF and mitogen-activated protein kinase kinase (MEK) inhibitors. However, most patients that initially respond to therapies develop drug resistance within months. Acquired resistance to targeted therapies can be due to additional genetic alterations in melanoma cells and to non-genetic events frequently associated with transcriptional reprogramming and a dedifferentiated cell state. In this second scenario, it is possible to identify pro-fibrotic responses induced by targeted therapies that contribute to the alteration of the melanoma tumor microenvironment. A close interrelationship between chronic fibrosis and cancer has been established for several malignancies including breast and pancreatic cancers. In this context, the contribution of fibrosis to drug adaptation and therapy resistance in melanoma is rapidly emerging. In this review, we summarize recent evidence underlining the hallmarks of fibrotic diseases in drug-exposed and resistant melanoma, including increased remodeling of the extracellular matrix, enhanced actin cytoskeleton plasticity, high sensitivity to mechanical cues, and the establishment of an inflammatory microenvironment. We also discuss several potential therapeutic options for manipulating this fibrotic-like response to combat drug-resistant and invasive melanoma.

## 1. Introduction

Cancer is defined as a disease of chronic inflammation. Fibrosis, a pathological feature of chronic inflammatory diseases, is in fact known to predispose and enhance cancer initiation and progression, mimicking the mechanism of a “non-healing wound” [1]. In addition to cancer-induced chronic inflammation, a fibrotic-like microenvironment can also be induced by anti-cancer treatments, such as traditional chemotherapies and radiotherapy [2]. One common link between fibrosis and cancer is represented by myofibroblasts. As shown in several systems, a deregulated process of wound healing driven by myofibroblasts leads to the accumulation of scar tissue and consequently to tissue fibrosis [3]. On the other hand, cancer-associated fibroblasts [4], a stromal cell population of the tumor microenvironment with tumorigenic properties, behave in a way close to myofibroblasts in the process of wound healing [5]. The interrelation between fibrosis and cancer has been established for several kind of malignancies, including breast [6,7], pancreatic [8], and lung [9,10] cancers, as well as melanoma [11]. Importantly, in melanoma, not only local stromal fibroblasts but cancer cells themselves can acquire a myofibroblast-like phenotype characterized by a contractile phenotype [12]. 

In this review, we summarize recent studies that have identified profibrotic responses and the acquisition of hallmarks of fibrosis as a consequence of MAPK-targeted therapies for the treatment of melanoma. First, we give an overview of the origin of melanoma and its clinical management. We then describe the main functional properties of myofibroblasts in wound healing and fibrosis and how melanoma cells can highjack some of them under BRAF and MEK inhibitor treatment. Finally, we discuss potential therapeutic options to target this fibrotic-like response in the context of melanoma resistance.

## 2. Melanoma

Cutaneous melanoma is a deadly form of skin cancer, accounting for 80% of skin cancer-related deaths [13]. It originates from malignant transformation of melanocytes, which are pigment-producing cells developing from the neural crest. Melanin, the main pigment produced by melanocytes, is delivered to keratinocytes through melanosomes to protect their nucleus from ultraviolet (UV) radiation-induced DNA damage [14]. Melanoma development is influenced by genetic factors, including germline mutations of genes involved in skin pigmentation and cell-cycle control [15,16] or the activation of mutations in the MAPK/extracellular signal-regulated kinase (ERK) pathway [17]. On the other hand, skin exposure to UV radiation is recognized as a major environmental factor linked to melanomagenesis [14]. Melanoma development commonly begins with a benign proliferative lesion in which melanocytes eventually enter a senescent-like state to generate melanocytic naevi. Additional mutations impair tumor-suppressor genes such as phosphatase and tensin homolog (*PTEN*) and inactivate fail-safe pathways to bypass senescence, sustain proliferation, and drive the spread of malignant melanoma metastastic [14]. 

Surgical resection of early stage melanoma ensures excellent survival rates (98%). However, once disseminated, melanoma constitutes a real therapeutic challenge because of its heterogeneity and phenotypic plasticity. Genetic classification of melanoma defines four different subtypes. The first three include melanomas harboring *BRAF*, *NRAS,* or neurofibromin 1 (*NF1*) mutations, respectively, and show constitutive activation of the MAPK pathway. The fourth subgroup includes malignancies that are not classified in the first three groups [18]. Interestingly, an activating mutation of *RAC1* has been recently identified as driver in melanoma, opening new therapeutic avenues for treatment [19,20]. Cutaneous melanoma also appears as one of the most heterogeneous cancers because of its high mutational burden due to sun exposure [21] and the acquisition of epigenetic modifications that include chromatin remodeling, differential expression of non-coding RNAs, and changes in DNA methylation status [22]. 

Better understanding of the molecular alterations driving melanoma progression has allowed the development of therapies targeting the constitutively activated MAPK signaling cascade observed in the majority of melanomas. In particular, the combination of inhibitors of oncogenic BRAF^V600^ mutants (BRAFi) and MEK inhibitors (MEKi) achieves significant clinical responses in patients with *BRAF*-mutated melanoma [23,24]. In addition, the discovery of regulatory molecules of the immune system has paved the way to revolutionary therapies for melanoma, defined as “immunotherapies”. These therapies are based on monoclonal antibodies targeting immunomodulatory receptors such as cytotoxic T-lymphocyte associated protein 4 (CTLA-4) or programmed cell death protein 1 (PD-1) that modulate the activity of cytotoxic T cells, thereby triggering anti-tumor immune responses [25,26]. However, only 30% to 50% of patients respond to anti-PD1 in combination or not with anti-CTLA-4 and adverse side effects frequently lead to treatment failure [27,28].

Even if a combination of BRAF and MEK inhibitors shows an unparalleled response rate in melanoma, a large proportion of patients eventually relapse [29]. A decade of extensive investigations has identified multiple mechanisms of resistance to MAPK-targeted therapies, involving both genetic and non-genetic mechanisms (Figure 1). Analysis of tumors from relapsed patients reveals that in 70% of cases, resistance to mono-treatment with BRAFi is caused by the reactivation of the MAPK pathway in a BRAF-independent manner. Common genetic mechanisms leading to MAPK reactivation include *NRAS* overexpression, *NRAS* activating mutations, and the loss of the MAPK pathway negative regulator *NF1*, all these events acting upstream of BRAF. On the other hand, downstream of BRAF, overexpression or mutations of *MEK* triggers MAPK reactivation. Together with the reactivation of MAPK signaling, genetic alterations in the phosphoinositide 3-kinase (PI3K)–PTEN–AKT axis are responsible for relapse in 22% of patients. Overall, the genetic alterations identified in BRAFi mono-therapy-resistant tumors are also found in BRAFi/MEKi-resistant tumors [30].

Matched comparison of pre- and post-relapse tumors under MAPKi treatment shows that alone the acquisition of de novo genetic mutations is not able to explain the variety of resistance mechanisms observed in melanomas [31,32,33]. A major non-genetically-driven mechanism of drug resistance stems from melanoma cell plasticity (Figure 1). At least two distinct cell populations characterized by a “proliferative” differentiated melanocytic phenotype or by an “invasive” dedifferentiated mesenchymal phenotype have been initially identified in melanoma [34,35]. Phenotype reprogramming is driven by changes in the activity of melanocytic lineage master regulators. Traditionally, microphthalmia-associated transcription factor (MITF), the transcriptional master regulator of pigment production, is considered a marker of the proliferative phenotype, while the receptor tyrosine kinase (RTK) AXL is a marker for the invasive one [34,35,36]. These distinct subpopulations can fluently convert from one phenotype into another in response to external stimuli from the tumor stroma such as hypoxia, inflammation, and nutrient starvation [37,38,39]. Phenotype plasticity also plays a role in the adaptation of melanoma cells to MAPK-targeted therapies [40]. The initial phase of treatment is characterized by an increased percentage of MITF^high^ cells which provide a drug-resistant state [41]. In parallel, cell populations characterized by a dedifferentiated invasive signature, the upregulation of RTKs including platelet-derived growth factor receptor beta (PDGFRβ), epidermal growth factor receptor (EGFR), nerve growth factor receptor (NGFR), insulin-like growth factor receptor 1 (IGF1R), and AXL [31,33,42,43,44,45], and the loss of MITF and its upstream regulator SRY-box transcription factor 10 (SOX10) [33] co-emerge, with the exclusion of *NRAS* mutations [31]. As RTKs upregulation drives the activation of MAPK-independent survival pathways, RTKs^high^ and MITF^low^ melanoma cells are resistant to MAPK inhibition, and it has been proposed that dedifferentiated and slow cycling melanoma cells may constitute a reservoir of cells from which resistant cells can emerge through the acquisition of additional mutations [33,46,47]. These subpopulations show chromatin modifications as well as upregulation of histone demethylases [48,49], and their dedifferentiated state can be transient or stabilized by BRAFi treatment through differential methylation of tumor cell-intrinsic CpG sites and epigenetic reprogramming [32,50,51]. Recurrent upregulation of hepatocyte growth factor receptor (MET), downregulation of lymphoid enhancer binding factor 1 (LEF1), and the enrichment of the Yes-associated protein 1 (YAP1) signature were identified as drivers of the acquired resistance [32] (Figure 1). Importantly, MAPKi resistance is correlated in half of melanomas with intratumoral CD8 T-cell exhaustion, implicating the dedifferentiated cell state in cross-resistance to anti PD-1/programmed cell death 1 ligand 1 (PD-L1) immunotherapy [32]. BRAFi also act as non-canonical ligands for the transcription factor aryl hydrocarbon receptor (AhR) to maintain melanoma cells in a proliferative and drug sensitive state. Conversely, high canonical AhR activity mediates drug resistance through the activation of a dedifferentiated cell state, suggesting AhR transcription factors as additional drivers of melanoma relapse [52]. 

Recently, the traditional model of melanoma phenotype switching has been extended by single-cell analysis, which paved the way to a more sophisticated definition of the transcriptional reprogramming induced by targeted therapies. Rambow et al. [53] showed that the combination of BRAFi/MEKi treatment triggers a progressive dedifferentiation of melanoma cells that is reflected by the acquisition of four distinct subtype signatures identified in the minimal residual disease (MRD) phase and that recall the different stages of embryonic development. One subpopulation is characterized by high MITF activity, which leads to a differentiated and pigmented state. Another subpopulation of drug-exposed cells acquires a ‘‘starvation’’-like transcriptional program. On the other hand, downregulation of MITF and induction of dedifferentiation is typical of two states: The invasive and the neural crest stem cell (NCSC) state. This last subpopulation is identified as a key driver of resistance, as a result of de novo transcriptional reprogramming promoted by the nuclear receptor retinoid X receptor gamma (RXRG) [53]. Of note, these four drug-resistant subtypes are highly reminiscent of the four drug-resistant states identified by Tsoi et al. [54]. The MITF^low^/SOX10^low^/AXL^high^-invasive subpopulation is also highly similar to the one described by Hoek et al. in the phenotype switch model [34]. 

Overall, these studies reveal that the co-emergence of drug-resistant states is driven by adaptive and non-mutational events, and in agreement with the study by Su et al. 2017 [55], the establishment of these states is considered to be the result of Lamarckian induction. 

## 3. Myofibroblasts in Tissue Repair and Fibrosis

Differentiation of fibroblasts into myofibroblasts is commonly viewed as a key event in the process of wound healing and tissue repair. The high contractile force that is generated by myofibroblasts is of pivotal importance for physiological tissue remodeling [56,57,58]. General hallmarks of myofibroblasts include a contractile cytoskeleton, linked to a high responsiveness to mechanical stimuli from the microenvironment; the ability to secrete and remodel the extracellular matrix (ECM); invasive properties; and the regulation of the inflammatory response (Figure 2). The contractile function of myofibroblasts relies on the assembly of focal adhesions [59] linked to integrin- and protease-dependent remodeling of the ECM. Focal adhesions are generated by the intracellular tension exerted by the actomyosin cytoskeleton and allow the transmission of intracellular forces to the ECM [58,60]. In addition, focal adhesions constitute a scaffold for signaling molecules, playing a role in the conversion of mechanical into biochemical signals, a process called mechanotransduction [61], which involves the actin-binding coactivator of transcription myocardin related transcription factor A (MRTFA) [62] and the Hippo pathway transcriptional effector YAP1 [63,64]. Nuclear translocation of MRTFA upon globular actin polymerization induces serum response factor (SRF)-mediated transcription of genes that regulate actin dynamics [62]. MRTFA role as a molecular linker between mechanical cues and myofibroblasts activation has been shown in scleroderma [65], lung fibrosis [66,67], and in the fibrotic response to myocardial infarction [68]. The contractile nature of the myofibroblast cytoskeleton is also of crucial importance in the remodeling of the ECM. This, together with the feature of ECM synthesis and degradation make myofibroblasts the main regulators of connective tissue remodeling during the physiological process of tissue repair [69].

The activation of the transforming growth factor beta (TGFβ) pathway is a central event in the initiation of fibrotic diseases [70]. Moreover, platelet-derived growth factor (PDGF) isoforms act as potent mitogens for cells of mesenchymal origin, fueling the expansion of the myofibroblast pool during the pathogenesis of fibrosis [71]. Nevertheless, local fibroblasts exposure to pro-inflammatory cytokines, such as tumor necrosis factor alpha (TNFα) and interleukin-1β (IL-1β) that are produced by immune cells, promotes their activation [72] (Figure 2). However, recent studies have demonstrated that, in the absence of exogenous cytokines, the pathological ECM produced in idiopathic pulmonary fibrosis (IPF) induces the differentiation of local fibroblasts into activated myofibroblasts. The establishment of the fibrotic ECM triggers a profibrotic loop involving the downregulation of miR-29 [73], a negative regulator of fibrotic genes, and an increased stiffness able to activate YAP1. This, in turn, upregulates the deposition of ECM [74]. Importantly, increased stiffness primes mesenchymal progenitors to acquire a so-called “mechanical memory” through the upregulation of miR-21, a positive regulator of fibrosis [75]. This paves the way to the hypothesis that, in the absence of organ injuries, fibrosis progression may take place in a “fibrogenic niche”, in which the ECM in itself is considered as a driver of organ fibrosis [76]. 

Myofibroblasts are also endowed with invasive abilities that allow them to invade into the wound matrix to promote tissue repair. Myofibroblast invasive properties are also critically implicated in the tumor microenvironment. In squamous cell carcinoma, cancer associated fibroblasts (CAFs) are in fact known to localize to the leading edge of the invasive front and to remodel the ECM in order to create tracks for the collective migration of cancer cells. This process is triggered by Oncostatin, a member of the interleukin 6 (IL-6) family that signals through the receptor subunit GP130-IL6ST (interleukin 6 signal transducer) and janus kinase 1 (JAK1) to generate Rho-dependent actomyosin contractility [77].

In addition, myofibroblasts are considered to be inflammatory cells because of their ability to regulate the inflammatory response through the release of soluble mediators of inflammation such as cytokines and chemokines [78,79,80], and the expression of adhesion molecules involved in the recruitment of immune cells to the inflammation site [80,81]. Another mediator of inflammation involved in fibrosis is endothelin-1 (ET-1): an endogenous vasoconstrictor which can be produced in the fibrotic context by myofibroblasts and inflammatory cells [82]. ET-1 is one of the main mediators of the profibrotic effects induced by TGFβ, and it is able to differentiate healthy fibroblasts into myofibroblasts, participating to the exacerbation of the profibrotic positive loop that leads to fibrosis progression [83]. In the case of chronic injury, a sustained activation of myofibroblasts triggers a positive loop that perpetuates the cycle of injury and results in scar tissue deposition and organ fibrosis.

## 4. Therapy-Induced CAF in Melanoma Resistance

An increasing interest in the investigation of the tumor microenvironment as a source of drug resistance has risen in recent years. Stromal cells are known to reduce cancer cell sensitivity to drugs through the release of soluble growth and inflammatory factors, cell–cell contact, as well as through the deposition of a deregulated ECM, a series of processes responsible for the so-called environment-mediated drug resistance (EM-DR) [84]. Importantly, EM-DR can be promoted by cancer cells through the recruitment and/or activation of fibroblasts into CAFs that show hallmarks of fibrosis-associated myofibroblasts. In the context of melanoma, MAPK inhibitors are able to promote stromal remodeling and CAFs activation, thereby fostering a drug-tolerant microenvironment (Figure 3). Hepatocyte growth factor (HGF) secretion by CAFs activates the RTK MET and the MAPK and PI3K/AKT pathways in melanoma cells, defining HGF secretion by local fibroblasts as an innate mechanism of resistance. Indeed, patients with stromal HGF expression have poorer responses than patients lacking its expression [85]. Conceptually similar to the work of Straussman et al. is the work of Wilson et al. [86], which shows that the autocrine, stromal, or systemic production of RTKs ligands, including HGF, drives the activation of survival pathways that affect the response to BRAFi. Another important soluble factor released by stromal cells that participates in melanoma sensitivity to targeted therapies is the Wnt-antagonist, secreted frizzled related protein 2 (sFRP2), the secretion of which by aged fibroblasts from the melanoma microenvironment attenuates the melanoma response to reactive oxygen species (ROS)-induced DNA damage and targeted therapies [87]. In addition, autocrine production of TGFβ by melanoma cells under BRAFi treatment transforms local fibroblasts into myofibroblasts [88]. On the other hand, BRAFi also activates local fibroblasts through a paradoxical stimulation of the MAPK pathway that confers CAFs with the ability to deposit a fibronectin-enriched matrix leading to the activation of pro-survival pathways in melanoma cells through integrin β1, focal adhesion kinase (FAK), and Src signaling [88,89], suggesting that residual disease can be supported by factors deriving from deregulated and fibrotic-like ECM triggered by targeted therapies. Importantly, therapy-induced inflammation also appears as an important source of non-mutational changes driving drug resistance. The development of inflammatory niches, in which MAPK inhibition amplifies the release of IL1β by tumor-associated macrophages, mediates the production of a CXC chemokine receptor 2 (CXCR2)-driven secretome by fibroblasts, which in turn promotes melanoma cell survival [90]. Together, these studies show a reciprocal contribution from melanoma cells, immune cells, and activated fibroblasts in mediating therapeutic escape.

## 5. Therapy-Induced Fibrotic Reprogramming of Melanoma Cells

Tumor plasticity consists of a series of genetic events and signaling adaptations that mediate escape from therapies. In recent years, the effect of targeted therapies on the contribution of melanoma cells to the fibrotic rewiring of the tumor microenvironment has been recognized. In addition, several studies support the notion that MAPK inhibitor treatment in *BRAF* mutant melanoma actually promotes the reprogramming of melanoma cells towards a CAF/myofibroblast-like phenotype that is a source of drug resistance and tumor progression (Figure 3). The study from Fedorenko et al. [91] showed that *PTEN*-null melanoma cells, after short-term BRAF inhibition, display perturbation in fibronectin-mediated adhesion signaling. BRAF inhibition in fact induces the formation (by melanoma cells) of a fibronectin-derived protective niche that activates signaling from α5β1 integrin/PI3K/AKT leading to an increase in the expression of the pro-survival myeloid cell leukemia 1 (MCL1) protein that mediates therapeutic escape. Globally, these perturbations induced during the short-term adaptation to BRAF inhibition, allow a small population of cells to escape therapies through increased PI3K/MAPK signaling. This pool of cells will then acquire secondary mutations to sustain tumor growth despite the therapeutic treatment. Importantly, a connection between PTEN loss and an increased deposition of fibronectin has been evidenced also in other systems and it is a feature of pathological fibrotic states, pointing out the close connection between a fibrotic-like stroma and resistance to treatment.

In addition to fibronectin, another ECM structural protein whose production by melanoma cells is affected by inhibitors of the MAPK pathway is type I collagen [12,92]. Type I collagen deposition is increased in vitro and in vivo following BRAF or ERK inhibition and this increased production is just partially induced by the activation of the TGFβ pathway suggesting the involvement of another signaling pathway in the upregulation of collagen by MAPKi. Consistently, the administration of MEKi alone or in combination with BRAFi increases ECM deposition and the formation of bundled collagen in a progressive way from the early stage to the late stage of treatment, with a marked dependency on bundled collagen for survival during the early stage of treatment [93]. Our recent study also shows that MITF^low^/AXL^high^ BRAFi-resistant cells exhibits a phenotype that is similar to that of CAFs, especially regarding ECM deposition and remodeling. The acquisition of CAF properties allows BRAFi-resistant cells to autonomously deposit a fibrillar ECM network, constituted of collagen fibers, collagen cross-linking enzymes, fibronectin, tenascin C, and thrombospondin 1, which in turn increases tolerance of naive melanoma cells to BRAFi and/or MEKi [12]. Most importantly, short-term treatment of naive melanoma cells with MAPK pathway inhibitors also triggers the autocrine production of an anisotropically aligned ECM enriched in collagen fibers and fibronectin. It also fosters the acquisition of an auto-amplifying CAF-like phenotype characterized by increased YAP1- and MRTFA-dependent mechanophenotype promoting tumor stiffening upon BRAFi treatment [12]. Consistently, the YAP1 signature has been identified as a driver event of melanoma-acquired resistance [32]. These studies underline the ability of MAPK-targeted therapies to biomechanically reprogram melanoma cells towards a CAF-like cell state that confers them with the ability to autonomously create, through altered ECM deposition and stiffening, a “safe-haven” that may promote drug resistance. In addition, collagen stiffening can promote melanoma differentiation via YAP/paired box 3 (PAX3)-mediated MITF expression [94], supporting the notion that collagen density and rigidity may also govern melanoma cell plasticity and intra-tumor heterogeneity. More insights into the microenvironment remodeling abilities conferred by MAPK-targeted therapies are provided by Sandri et al. [95], who identified an increased matrix metalloproteinase-2 (MMP2) activity in BRAFi-resistant cells responsible for a higher invasive index in resistant cells and collagen fibers remodeling. 

Reprogramming of melanoma cells toward a CAF/myofibroblast-like phenotype is also shown by the cytoskeletal features acquired by drug-treated melanoma cells and by an increased plasticity of the actin cytoskeleton. Early investigations into the role of oncogenic BRAF in the regulation of actin dynamics have shown that hyper-activation of the MAPK pathway disrupts cytoskeleton organization and focal adhesion formation through Rho GTPases signaling. Conversely, MEK inhibition or BRAF knockdown increases actin stress fiber formation and stabilizes focal adhesion dynamics through the downregulation of the Rho/Rho-associated protein kinase 1 (ROCK1) signaling antagonist Rnd3 [96]. A wider and more comprehensive approach to identify molecular adaptations to BRAF inhibition is taken in the study of Smit et al. [97]. Phosphoproteomics and genomics tools are used to identify drug targets that can sensitize melanoma cells to BRAF inhibition. ROCK1, a key regulator of actin cytoskeleton, is identified as a potential drug target to overcome adaptive or acquired resistance to BRAF inhibition. Several other studies indicate cytoskeleton rearrangements as the main driver of network rewiring following MAPK inhibition. High-resolution mass spectrometry identifies massive changes in the phosphoproteome of *BRAF* mutant melanoma cells after the acquisition of drug resistance. Importantly, the majority of these are related to key regulatory sites that control actin and microtubule dynamics, with a particular enrichment of factors belonging to the Rho/ROCK signaling pathway, identified here as a pivotal driver of plasticity and phenotypic transition [98]. Consistently, the acquisition of drug resistance through a dedifferentiated mesenchymal RTKs^high^ and MITF^low^ cell state is associated with extensive alterations in cell adhesion and actin cytoskeleton remodeling [12,99], as well as ROCK-dependent cell contractility [12]. 

In addition to its role as structural support to maintain cell shape, division, and migration, the actomyosin network transforms mechanical forces generated by microenvironment stiffness into biochemical signals that play a role in tumor progression and affect the sensitivity of cancer cells to chemotherapeutic agents. A similar scenario has been shown in melanoma where mechanosensitivity plays a role in the acquisition of resistance to MAPK-targeted therapies [12,89,100]. Key mediators in the translation of mechanical stimuli and cytoskeletal tension into transcriptional programs are the mechanotransducers YAP1 and TAZ (transcriptional co-activator with PDZ-binding domain). As demonstrated by [100], BRAFi treatment induces changes in the expression of actin cytoskeleton regulators through epigenetic mechanisms. In turn, perturbation of actin regulators triggers a deep cytoskeleton remodeling represented by an increase in the content of stress fibers. Together with YAP1/TAZ, MRTFA is another central mediator of mechanical stimuli also involved in the acquisition of MAPKi resistance in melanoma cells [12]. The role of MRTFA in resistance has been especially studied in MITF^low^/RTK^high^-resistant melanoma cells that acquire key features of CAFs, such as ECM remodeling activities. Our study shows that MAPK pathway inhibition confers melanoma cells with the ability to produce a rigid ECM in an autocrine way that modulates mechanosensing pathways involved in tumor stiffening. As a consequence of mechanical stress, YAP1 and MRTFA are translocated to the nucleus where they contribute to the ECM-mediated resistance to MAPK inhibitors, fueling a positive feedback loop between ECM deposition and mechanosensing, which is reminiscent of the myofibroblast-mediated fibrotic loop observed in fibrosis [76]. Activation of the mechanotransduction pathways is typical not only of acquired resistance but also of early adaptation to MAPK inhibition in vitro and in vivo. Thus, the mechanical adaptation of melanoma cells to BRAF inhibition may generate, in the long run, a pool of AXL^high^-resistant cells [12]. Moreover, combined treatment with BRAFi and the YAP1 inhibitor Verteporfin reduces tumor growth in vivo, confirming YAP1 as an important resistance factor in melanoma [12]. Similarly, high levels of RhoA signaling, coupled with elevated activation of MRTFA and YAP1, promotes BRAFi resistance in dedifferentiated melanoma cell lines characterized by a decreased expression of melanocyte lineage genes. Inhibition of the RhoA transcriptional program through ROCK inhibitor treatment re-sensitizes dedifferentiated melanoma cells to BRAFi treatment in vitro [101]. Significantly, enrichment of the YAP1 and ECM gene signature is also found in clinical melanoma specimens, suggesting the possible application of YAP1 or ROCK inhibition together with MAPK inhibitors in preventing the onset of resistance [12,97,101].

The importance of MRTFA as a resistance factor has been also investigated in *RAC1*^P29S^-mutated cells, the third most common mutation in melanoma after *BRAF^V600E^* and *NRAS^Q61^*. Constitutive activation of RAC1^P29S^ activates the MRTF/SRF transcriptional program, which leads in turn to a melanocytic to mesenchymal phenotypic switch [102]. Hence, *RAC1^P29S^*-mutated melanoma cells, characterized by a dedifferentiated phenotype, may constitute a reservoir of progenitor-like cells with reduced sensitivity to apoptosis, from which a tumor can relapse. Interestingly, resistance to BRAFi is reversed by co-treatment with a SRF/MRTF inhibitor, thus representing an interesting alternative to RAC1 inhibitors, which have to date demonstrated poor clinical success rates. Thus, cytoskeletal and mechanical adaptations that take place early under MAPK pathway inhibition confer a survival advantage to melanoma cells but also vulnerabilities that can be exploited to identify new druggable targets. In line with this notion, a recent study shows that myosin II activity and the ROCK pathway act as important survival factors that confer resistance to targeted therapy and to immunotherapy. Inhibition of myosin II causes the induction of lethal ROS and a loss of pro-survival signaling, which consequently trigger cell-cycle arrest and cell death [103]. As a consequence of the perturbation of the pathways related to cytoskeleton remodeling, melanoma cells also enhance their invasive abilities following MAPK pathway inhibition. In particular, Src family kinase (SFK) activation following MEKi administration increases integrin signaling that can be co-targeted with a MEKi and the SFKs inhibitor Sarcatinib [104]. The rewiring of pathways involved in actin cytoskeleton-dependent invasiveness is also described in the work of [105], in which the SFKs–FAK–signal transducer and activator of transcription 3 (STAT3) signaling axis is activated after BRAFi or MEKi treatment, leading to an invasive phenotype of melanoma cells. Moreover, SFKs participate in an EGFR–STAT3 axis involved in cytoskeleton remodeling and invasiveness of BRAFi-resistant melanoma cells, pathological outcomes that can be overcome with a combination of BRAFi and either Dasatinib or EGFR inhibitor [43]. Together, these studies outline the paradoxical effect of MAPK-targeted therapies in reprogramming melanomas towards a fibrotic-like resistant state.

## 6. Therapy-Induced Inflammation

Therapy-induced inflammation is also an important source of phenotype plasticity for melanoma cells and relapse. In the tumor stroma, MAPK inhibition enhances the recruitment of tumor-associated macrophages that, through TNFα release, increase the expression of MITF in melanoma cells. This transcription factor contributes to survival signaling through the expression of antiapoptotic genes. A combination of MAPK pathway inhibition with IκB kinase (IKK) inhibitors improves the therapeutic response, diminishing MITF expression in melanoma cells and blocking TNFα activity in tumor stroma [37]. In line with this, inflammation-induced melanoma cell dedifferentiation is linked to immunotherapy resistance in mice [106]. In this context, it is interesting to note that during the response phase of melanoma to BRAFi treatment, the induction of the pro-inflammatory and lung fibrosis factor ET-1 by MITF was described as a master mechanism regulating phenotypic heterogeneity and as a druggable target in the context of melanoma resistance [107]. MITF-induced secretome under BRAFi treatment includes the secretion of ET-1, which supports tumor growth by reactivating the ERK pathway in a paracrine manner. This pro-survival effect is observed in MITF^high^ and AXL^high^ melanoma subpopulations through endothelin receptor A (EDNRA) and endothelin receptor B (EDNRB) signaling, respectively [107]. The administration of EDNR antagonists [107] or antibody–drug conjugate targeting EDNRB [108] shows a beneficial effect in combination with MAPK inhibitors. Interestingly, a subset of resistant *BRAF* mutant melanoma cells shows enrichment in the signatures related to inflammation and nuclear factor-kappa B (NF-κB) signaling [32]. Consistently, we have demonstrated that dedifferentiated melanoma cells express inflammation-related genes such as pentraxin 3 (PTX3), which contribute to melanoma invasiveness and the mesenchymal-resistant phenotype via a Toll-like receptor 4 (TLR4)-NF-κB-TWIST pathway [109]. Interestingly, between the four drug-resistant states identified by Tsoi et al., undifferentiated subtypes (invasive and neural crest-like) are enriched for genes related to inflammation and show a higher recruitment of myeloid cells that support tumor growth and immunosuppression [54,110].

## 7. Translational Potential of Anti-Fibrotic Agents for Melanoma Therapy

The impact that the fibrotic-like phenotype has on melanoma behavior as a tumor promoting force and on the acquisition of resistance to MAPK-targeted therapies paves the way to the development of novel combinatorial therapeutic strategies. Given the significant overlapping in pathways involved in fibrosis and cancer (Figure 4), we discuss here the potential translational benefit of anti-fibrotic agents to delay and/or overcome resistance to targeted therapies in melanoma.

Nintedanib (BIBF1120), an inhibitor of PDGFR, vascular endothelial growth factor receptor (VEGFR), and fibroblast growth factor receptor 1 (FGFR1), was initially studied for its role in angiogenesis inhibition, but its importance in the treatment of fibrotic disease derives from the ability to suppress myofibroblast differentiation and to reduce collagen deposition [111]. Recent clinical trials have shown the efficacy and tolerability of Nintedanib in lung fibrosis treatment [112,113]. Moreover, it has been shown that in combination with traditional chemotherapy, BIBF1120 improves clinical outcomes in terms of response rate and progression-free survival in non-small cell lung cancer patients [114,115]. Conversely, another multi-kinase inhibitor directed against PDGFR and approved for the treatment of hepatocellular and renal carcinoma, Sorafenib, has shown anti-fibrotic activity on liver fibrosis in preclinical models [116]. Furthermore, Imatinib, in addition to its clinical application for chronic myeloid leukemia, exerts therapeutic efficacy in the treatment of gastrointestinal stromal tumors [117] and in nephrogenic systemic fibrosis [118] thanks to its ability to inhibit the PDGF receptor and c-KIT. Accordingly, the new generation of breakpoint cluster region (BCR)-ABL inhibitors (Dasatinib and Nilotinib) is currently being used for the treatment of systemic sclerosis [119,120,121].

A key factor that connects fibrosis to cancer is the pleiotropic cytokine TGFβ. Pirfenidone, a compound able to inhibit TGFβ signaling by preventing SMAD2/3 nuclear translocation, blocks ECM accumulation and myofibroblast proliferation in vitro [122,123]. Recently, it has been approved for the clinical treatment of lung fibrosis [113] and tested in combination with chemotherapeutic compounds for the treatment of lung malignancies. As collagen is the main component of ECM, the most exploited approach to reduce its synthesis has been the inhibition of TGFβ signaling, which plays a regulatory role in collagen production. Among the different strategies aimed at the impairment of collagen synthesis, Halofuginone has shown efficacy in vitro and in vivo [124]. 

Targeting ECM-remodeling enzymes to prevent the disruption of ECM homeostasis has also become an attractive approach both in cancer therapy and fibrosis. Regulation of collagen cross-linking is mainly mediated by enzymes of the lysyl oxidase (LOX) family, which are upregulated by BRAFi treatment [12]. The LOX inhibitor β-aminopropionitrile (BAPN) is efficient in reducing collagen cross-linking and fibrotic scarring [125], but unfortunately, clinical trials have been halted due to drug toxicity. However, LOX and lysyl oxidase-like 2 (LOXL2) inhibition seem to be promising in cancer therapy, as reducing their activity decreased mechanotransduction in vitro and reduced tumor growth [126,127,128].

Integrins, a family of transmembrane receptors that mediate cell–matrix and cell–cell interactions, have been identified as participating in the fibrotic process and their knockdown dampens disease progression [129]. Because of their implication in the acquisition of therapy resistance in melanoma, therapies based on their inhibition, or the inhibition of their downstream signal transducer FAK, have a wide potential not only as anti-fibrotic [130,131] but also anti-cancer therapies. In this second scenario, FAK inhibition normalizes the fibrotic tumor microenvironment of pancreatic cancer and increases immune surveillance, improving the efficacy of immunotherapies [132]. 

YAP/TAZ signaling is also viewed as a molecular link between fibrosis and cancer [133]. The YAP1 inhibitor Verteporfin is efficient in preclinical models of kidney fibrosis [134] and it has been exploited as a molecular target in a pre-clinical model of melanoma, where it prevents the fibrotic phenotype induced by oncogenic BRAF inhibition [12]. YAP1 is also a core mediator of integrin β1 signaling in liver fibrosis. In this context, pharmacological inhibition of either pathway in vivo attenuates liver fibrosis and suggests a synergistic effect in the combined inhibition of integrins and the mechanosensor YAP1 [135]. Another critical regulator that links mechanical cues to aberrant remodeling of the extracellular matrix in fibrosis is MRTF. Anti-fibrotic agents inhibiting Rho/MRTF/SRF-mediated gene transcription significantly impair the development of bleomycin-induced dermal fibrosis in vivo [136] and decrease the activation of pancreatic stellate cells in the tumor microenvironment, ameliorating the possibilities of therapeutic intervention [137]. Rho/MRTF signaling is not only involved in the fibrogenic process but also in the aggressive phenotype of metastatic melanoma. Hence, targeting the MRTF transcriptional pathway appears as a novel approach for melanoma therapeutics [102,138]. Finally, an additional hallmark of tissue response to injury is the reorganization of actin cytoskeleton. The ROCK family of serine/threonine kinases orchestrates this process and it has been shown to contribute to the pathogenesis of a wide range of fibrotic diseases [139]. Consistently, ROCK inhibition has a huge potential in tackling the non-genetic mechanism of resistance in melanoma [97,103].

Together, these studies highlight the vast potential of anti-fibrotic drugs in combination with BRAF^V600^-targeted therapies for the development of original therapeutic approaches in melanoma.

## 8. Conclusions 

Cancer cell plasticity and adaptation to stressful environments appear as critical features during the development of therapeutic resistance and clinical relapses. Herein, we reviewed the paradoxical fibro-mechanic reprogramming of *BRAF*-mutant melanomas, which is achieved in response to MAPK pathway inhibition. In particular, the acquisition of this therapy-induced fibrotic-like phenotype, which seems quite unique to cutaneous melanoma, endows cancer cells with cell-autonomous abilities to resist treatments and escape challenging tumor microenvironments. Most importantly, therapy-induced reprogramming of the melanoma microenvironment may foster the establishment of tissue-specific malignant fibrogenic niches involved in tumoral heterogeneity and therapeutic escape. On the other hand, such non-genetic mechanisms also unveil novel vulnerabilities and opportunities for the development of fibrosis-oriented therapeutic strategies against refractory melanoma. 

## Figures and Tables

**Figure 1 cancers-12-01364-f001:**
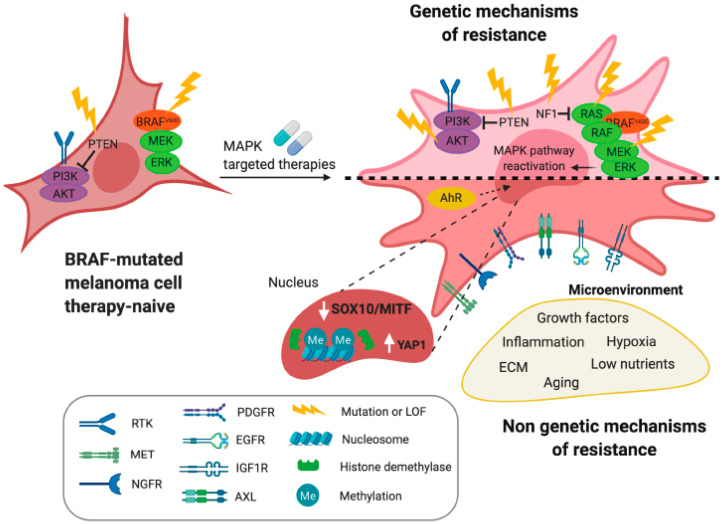
Mechanisms of resistance to mitogen-activated protein kinase (MAPK)-targeted therapies in melanoma. Melanoma cells evade targeted therapies by acquiring additional genetic alterations or through non-genetic mechanisms. LOF: loss of function; MEK: mitogen-activated protein kinase kinase; ERK: extracellular signal-regulated kinase; NF1: neurofibromatosis 1; RTK: receptor tyrosine kinase; MET: hepatocyte growth factor receptor; NGFR: nerve growth factor receptor; PDGFR: platelet derived growth factor receptor; EGFR: epidermal growth factor receptor; IGF1R: insulin-like growth factor 1 receptor; AXL: AXL receptor tyrosine kinase; PTEN: phosphatase and Tensin homolog; PI3K: phosphoinositide 3-kinase; MITF: microphthalmia-associated transcription factor; SOX10: SRY-box transcription Factor 10 (SOX10); YAP1: Yes-associated protein 1; AhR: aryl hydrocarbon receptor.

**Figure 2 cancers-12-01364-f002:**
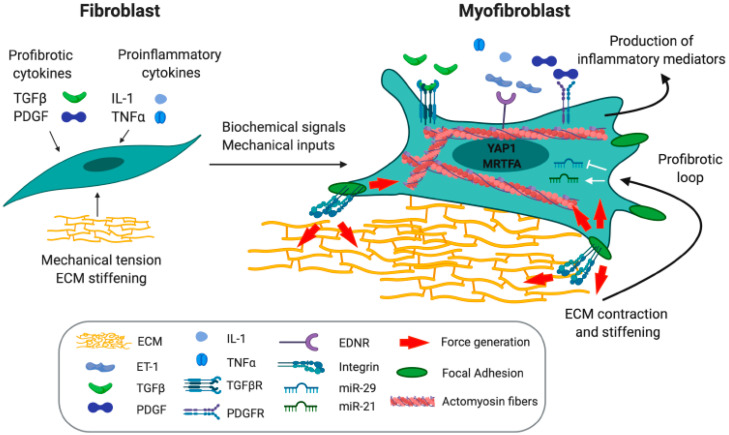
Fibroblast to myofibroblast transition. The differentiation of fibroblast to myofibroblast, which takes place during the physiological process of wound healing, leads to the pathogenesis of fibrotic diseases when deregulated. ET-1: endothelin 1; TGFβ: transforming growth factor beta; PDGF: platelet-derived growth factor; IL-1: interleukin 1; TNFα: tumor necrosis factor alpha; TGFβR: transforming growth factor beta receptor; PDGFR: platelet-derived growth factor receptor; EDNR: endothelin receptor; ECM: extracellular matrix; YAP1: Yes-associated protein 1; MRTFA: myocardin-related transcription factor A.

**Figure 3 cancers-12-01364-f003:**
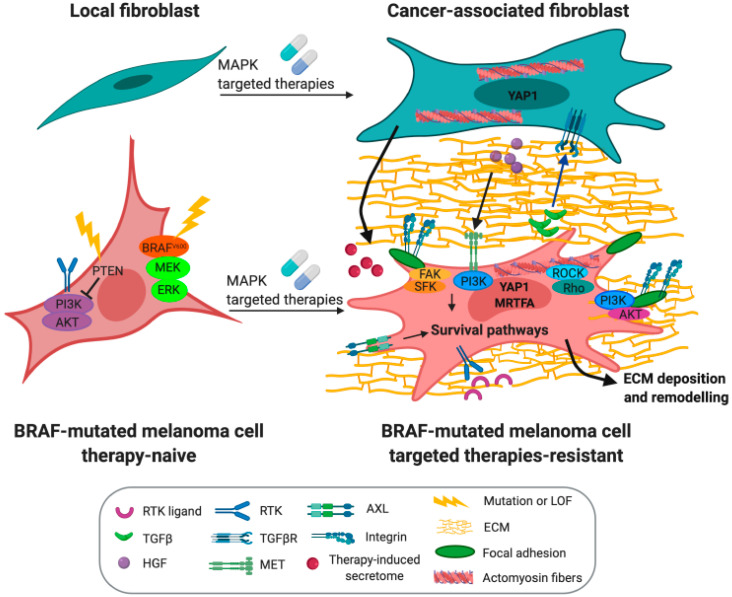
MAPK pathway inhibition mediates tumor microenvironment remodeling as a source of therapy resistance. Melanoma cells and cancer-associated fibroblasts cross-talk mediates therapeutic escape from MAPK-targeted therapies. MEK: mitogen-activated protein kinase kinase; ERK: extracellular signal-regulated kinase; RTK: receptor tyrosine kinase; TGFβ: transforming growth factor β; HGF: hepatocyte growth factor; TGFβR: transforming growth factor beta receptor; MET: hepatocyte growth factor receptor; AXL: AXL receptor tyrosine kinase; ECM: extracellular matrix; FAK: focal adhesion kinase; PTEN: phosphatase and Tensin homolog; PI3K: phosphoinositide 3-kinase; SFK: Src family kinase; ROCK: Rho-associated protein kinase; YAP1: Yes-associated protein 1; MRTFA: myocardin-related transcription factor A.

**Figure 4 cancers-12-01364-f004:**
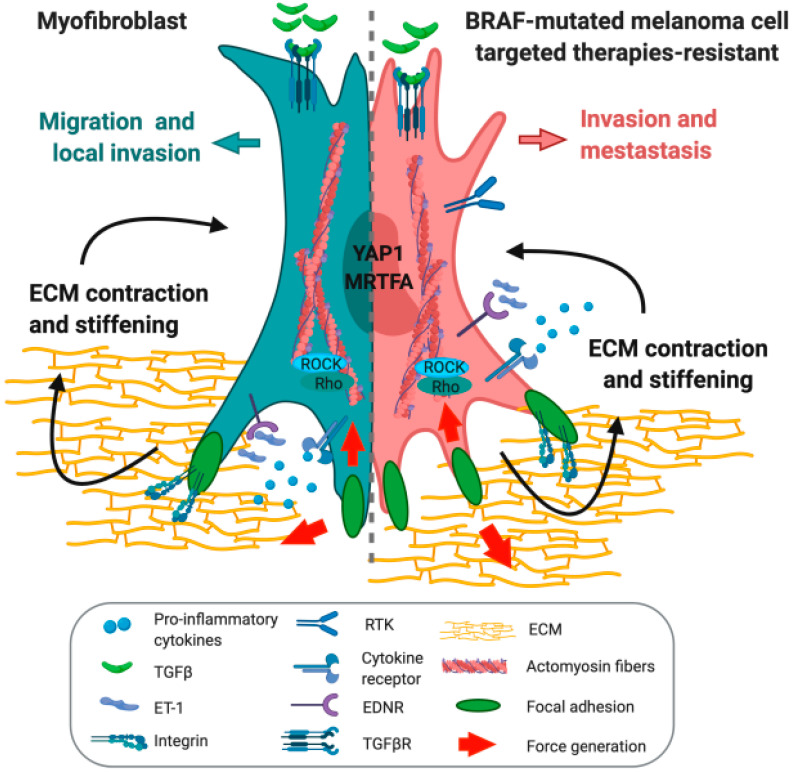
Comparison of the features of a myofibroblast and a targeted therapy-resistant melanoma cell. Schematic depiction of the reprogramming of melanoma cells toward a myofibroblast-like phenotype. TGFβ: transforming growth factor beta; ET-1: endothelin 1; RTK: receptor tyrosine kinase; EDNR: endothelin receptor; TGFβR: transforming growth factor beta receptor; ECM: extracellular matrix; YAP1: Yes-associated protein 1; MRTFA: myocardin-related transcription factor A; ROCK: Rho-associated protein kinase.

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
