# Peer review of "Bad Neighborhood: Fibrotic Stroma as a New Player in Melanoma Resistance to Targeted Therapies"

_cancers, 2020, doi:10.3390/cancers12061364_

Round 1
Reviewer 1 Report
The present work addresses the role of fibrosis in determining resistance to BRAF-MEK targeted therapy in metastatic melanoma. Although the authors have done a thorough review, evidences in this field are limited and this issue is not worthy for a review work. The scientific resonance is poor.
Author Response
Reviewer 1
The present work addresses the role of fibrosis in determining resistance to BRAF-MEK targeted therapy in metastatic melanoma. Although the authors have done a thorough review, evidences in this field are limited and this issue is not worthy for a review work. The scientific resonance is poor.
Reply: We are very sorry that we could not convince the reviewer that there is actually plenty of scientific evidence to justify a review work about the relevance of a fibrotic microenvironment in regards to melanoma therapeutic response. Microenvironmental factors such as TGFb, growth factors, inflammatory mediators, ECM accumulation and stroma stiffening, and the presence of myofibroblast-like cells, which are hallmarks of fibrotic-like diseases, have been clearly associated to melanoma response to targeted therapies and non-genetic acquired mechanisms of resistance (as reviewed in our manuscript). Recent evidence also indicate that resistance develops when tumor cells migrate and expand in a stroma reminiscent of a fibrotic microenvironment. In addition, it is known that desmoplastic melanomas exhibit a fibrotic and collagen-rich stroma, which impacts on therapies.
Importantly, melanoma cells exposed to therapies against the BRAF oncogenic pathway can reprogram towards a mesenchymal undedifferentiated cell state, which exhibit several traits of myofibroblasts or cancer-associated fibroblasts. We and others have shown that this therapy-induced cell state contributes to a fibrotic microenvironment associated to drug resistance. Together, these data highlight the therapeutic potential of anti-fibrotic agents to combat melanomas as we reviewed in our manuscript.
In order to improve it, our manuscript has been edited for English language and style.
We sincerely hope that the above arguments will convince the reviewer about the scientific resonance and quality of our review work.
Reviewer 2 Report
Very well designed review.
Thorough literature review.
Concise but clear summary of this rather novel and complex area of research
No language issues.
Minor remarks:
- line 82: 30-50% of patients respond to anti-PD1 +/- anti-CTLA4
- line 120: for the sake of clarity, I would employ drug-sensible or drug-resistant instead of drug-tolerant
Author Response
Reviewer 2
Comments and Suggestions for Authors
Very well designed review.
Thorough literature review.
Concise but clear summary of this rather novel and complex area of research.
No language issues.
Reply: We thank the reviewer for his/her interest about our review, and for his/her supportive remarks.
Minor remarks:
- line 82: 30-50% of patients respond to anti-PD1 +/- anti-CTLA4
We thank the reviewer for this pertinent remark. We have now modified the sentence to include the percentage of melanoma patients, which are responding to immunotherapies. (lines 139 to 141 of the revised manuscript)
- line 120: for the sake of clarity, I would employ drug-sensible or drug-resistant instead of drug-tolerant
To meet the reviewer’s recommendation, we have replaced drug-tolerant (cells, states, subtypes) by drug-resistant (cells, states, subtypes).
(lines 405, 438, 439, 442, and 1293 of the revised manuscript)
In order to improve it, our manuscript has been edited for English language and style, and minor spell check has been performed.
We hope that the reviewer will now be satisfied with the revised manuscript.
Reviewer 3 Report
- Factually, I think this is a fine and interesting read; it makes some interesting insights into potential drugs to treat melanoma progression/drug resistance by targeting fibrosis. I have no recommendations for alterations to the text or reference list
- The most common errors are typos and grammar
- Awkward/unnecessary pluralization or word choice
- Occasional subject-verb disagreement
- Some awkward sentence structure, some bordering on run-on sentences
- General omission of the Oxford comma, but this is just my stylistic preference
- There are a few Greek symbols missing like after IL1 and TGF in some spots
- A few parenthetical abbreviations need to be added, e.g. aryl hydrocarbon receptor (AhR)
- Generally, I think one more proofreading would be beneficial, but these errors do not change the general meaning/accuracy of the review
Author Response
Reviewer 3
Comments and Suggestions for Authors
- Factually, I think this is a fine and interesting read; it makes some interesting insights into potential drugs to treat melanoma progression/drug resistance by targeting fibrosis. I have no recommendations for alterations to the text or reference list
Reply: We thank the reviewer for his/her interest about our review, and for his/her supportive comments.
- The most common errors are typos and grammar
- Awkward/unnecessary pluralization or word choice
- Occasional subject-verb disagreement
- Some awkward sentence structure, some bordering on run-on sentences
- General omission of the Oxford comma, but this is just my stylistic preference
In order to improve it, our manuscript has been edited for English language and style, and spell check has been performed. The omission of the Oxford comma has been corrected.
- There are a few Greek symbols missing like after IL1 and TGF in some spots
The missing Greek symbols have been added in the revised manuscript.
- A few parenthetical abbreviations need to be added, e.g. aryl hydrocarbon receptor (AhR)
The missing parenthetical abbreviations have been added in the revised manuscript.
- Generally, I think one more proofreading would be beneficial, but these errors do not change the general meaning/accuracy of the review
Following the reviewer’s recommendation, our manuscript has been edited for English language and style. We also performed a careful proofreading of the manuscript in order to correct orthographic and stylistic errors.
We hope that the reviewer will now be satisfied with the revised manuscript.